# Brain Metastasis Segmentation Network Trained with Robustness to Annotations with Multiple False Negatives

**Darvin Yi**[1]                                                DARVINYI@STANFORD.EDU
[1] *Department of Biomedical Data Science at Stanford University, Stanford, CA 94305 USA*

**Endre Grøvik**[2]                                          ENDRE.GROVIK@MN.UIO.NO
[2] *Department for Diagnostic Physics at Oslo University Hospital, Oslo, Norway*

**Michael Iv**[3]                                                  MIV@STANFORD.EDU
[3] *Department of Radiology at Stanford University*

**Elizabeth Tong**[3]                                           ETONG@STANFORD.EDU

**Greg Zaharchuk**[3]                                          GREGZ@STANFORD.EDU

**Daniel Rubin**[1,3]                                           RUBIN@STANFORD.EDU

**Editors:** Under Review for MIDL 2020

## Abstract

Deep learning has proven to be an essential tool for medical image analysis. However, the need for accurately labeled input data, often requiring time- and labor-intensive annotation by experts, is a major limitation to the use of deep learning. One solution to this challenge is to allow for use of coarse or noisy labels, which could permit more efficient and scalable labeling of images. In this work, we develop a lopsided loss function based on entropy regularization that assumes the existence of a nontrivial false negative rate in the target annotations. Starting with a carefully annotated brain metastasis lesion dataset, we simulate data with false negatives by (1) randomly censoring the annotated lesions and (2) systematically censoring the smallest lesions. The latter better models true physician error because smaller lesions are harder to notice than the larger ones. Even with a simulated false negative rate as high as 50%, applying our loss function to randomly censored data preserves maximum sensitivity at 97% of the baseline with uncensored training data, compared to just 10% for a standard loss function. For the size-based censorship, performance is restored from 17% with the current standard to 88% with our lopsided bootstrap loss. Our work will enable more efficient scaling of the image labeling process, in parallel with other approaches on creating more efficient user interfaces and tools for annotation.

**Keywords:** Brain Metastasis,Segmentation,Deep Learning,False Negative,Noisy Label

## 1. Introduction

In recent years, deep learning has advanced many areas of computer vision, such as image classification (Krizhevsky et al., 2012; Simonyan and Zisserman, 2014; Szegedy et al., 2016; He et al., 2016; Huang et al., 2017), object detection (Ren et al., 2015; He et al., 2017; Redmon and Farhadi, 2017; Lin et al., 2017), and semantic segmentation (Long et al., 2015; Ronneberger et al., 2015; Badrinarayanan et al., 2017; He et al., 2015; Chen et al., 2017a,b). This has also led to an explosion in high-profile applications of deep learning in medical image analysis (Ting et al., 2017; Esteva et al., 2017). Datasets in medical imaging are starting to achieve comparable sizes as those for classification-level labels; examples include

CheXNet (Rajpurkar et al., 2017b), MURA (Rajpurkar et al., 2017a), and DREAM Digital Mammography Challenge. However, datasets for dense prediction tasks like segmentation remain limited, comprising at most hundreds of patients (Heller et al., 2019; Bakas et al., 2018; Pedrosa et al., 2019; Bilic et al., 2019).

Current approaches to facilitate efficient generation of labeled data include improving management and sharing of expert annotations on medical images (Rubin et al., 2008; Moreira et al., 2015) and mining data automatically from the picture archiving and communication system (PACS) (Yan et al., 2018; Shin et al., 2015). However, these methods generally involve processing data on physician interactions in a given database system, often requiring specialized code unique to each institution or database manufacturer. Others have examined the potential for crowdsourcing labels (Albarqouni et al., 2016; Irshad et al., 2014), as done for many computer vision datasets, but medical images often require extensive domain knowledge to achieve accurate annotations. In this work, we focus not on increasing the quality or quantity of available data but rather on improving methods for learning from these data. Specifically, we develop a network that is highly robust to noisy labels, especially false negatives (FNs), to reduce the requirements for large amounts of fine and dense annotations.

We build on the work of Grøvik et al. (2020), using brain metastasis as a model system to evaluate the impact of FN annotations in a segmentation task. Previous work in this area, such as Lu et al. (2016), showed that weakly supervised learning involving super-pixel alignment could help fix noisy boundaries between labels, but such methods would not be adequate for cases where full segments were not labeled, as when an entire lesion was missed by an annotator. The weak label learner (WELL) (Sun et al., 2010) and multi-label with missing labels (MLML) (Wu et al., 2015) frameworks can deal with missing labels but not fully misclassified annotations. Our work builds primarily on the foundation of Reed et al. (2014), which trains a network on noisy (misclassified) labels. We expand the method from classification to segmentation and also modify the loss function under our assumption that FNs, not false positives (FPs), are the main source of noise.

We present the following novel contributions:

1. We have developed, to our knowledge, the first segmentation network dealing with whole-lesion FN labels.

2. We have created a "lopsided bootstrap loss" that assumes prevalence of annotations with FNs and recovers performance through entropy regularization.

3. We have demonstrated that this method preserves performance for exceptionally high induced FN rates (as much as 50%), where FN lesions are chosen either at random or based on size.

## 2. Data

This dataset, introduced in Grøvik et al. (2020), comprises 156 patients examined at Stanford Hospital with known brain metastases and no prior treatment (surgery or radiation). Our dataset is split 100/5/51 for training/validation/test. The use of the validation set is described in 3.3. The test set was chosen to have 17 patients each having 1-3 lesions, 4-10 lesions, and 10+ lesions, to ensure that our model was not biased for cases with numerous or sparse

lesions. For each, four MR pulse sequences are available: pre- and post-contrast T1-weighted 3D fast spin echo (CUBE), post-Gd T1-weighted 3D axial inversion recovery prepped fast spoiled gradient-echo (BRAVO), and 3D fluid-attenuated inversion recovery (FLAIR). The FLAIR as well as pre- and post-contrast CUBE series were co-registered to the post-contrast BRAVO series using the nordicICE software package (Nordic Neuro Lab, Bergen, Norway). The annotations were also done on the BRAVO series.

The metastasis lesions were labeled by two neuroradiologists with a combined experience of 13 years. Some annotations were made by a fourth-year medical student, which were then edited by the aforementioned neuroradiologists. The labeling was done on a professional version of OsiriX (Rosset et al., 2004) using the polygon tool, which requires the annotator to click once per vertex per slice per lesion. The average 3D lesion required 45 clicks to fully annotate. The annotation time depended on the number and size of lesions and was highly variable, ranging from about 1 minute for patients with a single large (greater than 1cm in diameter) lesion to 3 hours for patients with over a hundred smaller lesions. For maximal use of physician time in gathering more data, no cases were read by more than one reader. This design choice leaves our study's reference standard vulnerable to error.

## 3. Methods

### 3.1. Lesion Censoring

We simulate FN by performing lesion censoring, or deleting lesions in the expert annotations.

### Stochastic 50% Censoring

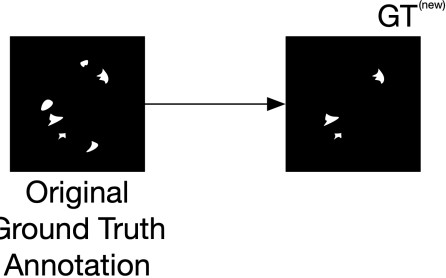

Figure 1: **A graphical example of the lesion censoring process.** The example shown here illustrates lesion censoring with a 50% stochastic censoring rate.

**Stochastic Censoring.** In stochastic censoring, we specify a rate $p$ of FNs to induce in the labeled data. Each lesion (or 3D CC) is then independently censored with probability $p$, yielding an overall FN rate of $p$ over all lesions in the training data. Since censoring is stochastic, there are exponentially many possible combinations ($2^n$, where $n$ is the number of lesions) for censored vs. retained lesions.

**Size-Based Censoring.** To better model clinician errors, we introduce size-based censoring. In this method, we systematically censor the smallest lesions (by volume) across all patients to achieve the desired FN rate of $p$, such that proportion $p$ of the total number

of lesions have been removed.This method generates the set of censored vs. retained lesions deterministically. While it better approximates physician error than does stochastic censoring, it does not capture other sources of error, such as lesions that are subtle due to lack of contrast with healthy tissue rather than small size.

For both stochastic and size-based censoring, we use a very harsh FN rate of 50%. While this rate may be too high to accurately simulate clinician error, this choice allows us to better evaluate the performance limit of our methods, as a method robust to such a high FN rate will likely accommodate the lower rate of typical clinical applications.

### 3.2. Bootstrap Loss

Here, we define the different loss functions with which we will train our network.

**Class-Based Loss Weighting.** The most naive approach to solving the FN problem is to introduce an additional weighting for positive cases. Since we consider the limit in which FNs represent the main source of annotation error, we can attempt to improve performance by upweighting the loss for incorrectly classifying the positive cases, as defined in equation 1.

$$\mathcal{L}(Y, \hat{Y}) = \left\{ \begin{array}{ll} CE(Y, \hat{Y}) & \text{if } Y == 0 \\ \alpha * CE(Y, \hat{Y}) & \text{if } Y == 1 \end{array} \right. \tag{1}$$

Specifically, we define a weighting parameter $\alpha$ which describes the multiplicative factor for the positive cases. For example, with $\alpha = 10$, positive case pixels will receive $10\times$ the weight of negative case pixels.

**Bootstrap Loss.** Based on (Reed et al., 2014), our work focuses on the bootstrap loss, as given in equation 2. This loss is a weighted average between the common cross entropy (CE) loss and a secondary CE loss between our predicted probabilities, $\hat{Y}$, and our predicted classification one-hot encodings, $\text{argmax}(\hat{Y})$. This creates a feedback loop, where we push predictions to further increase the "winning" logit value and lower the other losing values. Thus, this is a form of entropy minimization, and the bootstrap loss is a regularization of the entropy of our predicted probability distribution.

$$\mathcal{L}(Y, \hat{Y}) = \beta * CE(Y, \hat{Y}) + (1 - \beta) * CE(\text{argmax}(\hat{Y}), \hat{Y}) \tag{2}$$

We define a parameter $\beta \in (0, 1]$, which defines the proportion of loss represented by the classical cross entropy with the remainder being our bootstrap cross entropy. For example, $\beta = 1$ represents the baseline case where we only use the standard cross entropy. $\beta = 0.1$ would represent 10% of our loss coming from the CE between our predictions and our (potentially noisy) target annotations while 90% of our loss comes from the feedback loop of the bootstrap loss component. A diagram of the bootstrap loss can be found in figure 2.

**Lopsided Bootstrap Loss.** Given our assumption that errors are predominantly FNs, we can further improve performance by introducing a lopsided bootstrap loss that also incorporates class-based loss weighting. Taking our noisy target labels, we can separate the loss into two cases: (1) the target is positive and (2) the target is negative. When the target is positive, we will weight the loss by our $\alpha$ factor, as in the class-based loss weighting. In the case where the target is negative, we will apply the bootstrap loss with parameter $\beta$. With $\beta = 1$, this loss simply reduces to class-based loss weighting where positive cases are upweighted by $\alpha$.

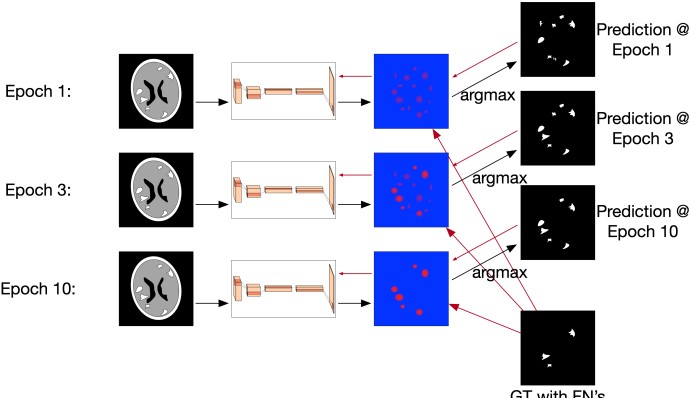

Figure 2: **Diagram of Bootstrap Loss** At every iteration of training, the loss is shared between treating the noisy annotations and our own binarized predictions as the target.

$$\mathcal{L}(Y, \hat{Y}) = \begin{cases} \beta * CE(Y, \hat{Y}) + (1 - \beta) * CE(\text{argmax}(\hat{Y}), \hat{Y})) & \text{if } Y == 0 \\ \alpha * CE(Y, \hat{Y}) & \text{if } Y == 1 \end{cases} \tag{3}$$

Thus, we introduce two hyperparameters $(\alpha, \beta)$ into our model search pipeline. We chose exponentially distant values of $\alpha$ and $\beta$ to limit the number of models we would have to train and evaluate: $\alpha \in \{3, 10, 30\}$ and $\beta \in \{1, 0.5, 0.1\}$.

### 3.3. Implementation Details

We follow methodology very similar to prior work on metastasis segmentation from Yi et al. (2019) and Grøvik et al. (2019). We use DeepLabv3 (Chen et al., 2017b) for the core segmentation network, but our lopsided bootstrap loss is generalizable to any architecture. Our network is a 2.5D network, with all 5 z-slices (with center slice being predicted upon) of all four pulse sequences being stacked in channel-space for our input. Our input tensor would be of size $256 \times 256 \times 20$ for each frame.

All code was written using PyTorch (Paszke et al., 2017). Networks were trained on a commercial-grade workstation with two NVIDIA 1080Ti GPUs. Networks were left to train for about 10 epochs, or approximately 24 hours. The network runs at about 100 ms per frame, corresponding to a runtime of about 30 s per 3D MR scan. The network with the best performance on the validation set was chosen for testing. The validation set had the same lesion censorship as the training set, whether we had no censorship, stochastic censorship, or size-based censorship. The test set had no censorship and used all lesions of the original annotation set.

The baseline model as reported in Yi et al. (2019) had extensive hyperparameter search, to ensure that the baseline approach had the strongest possible model. All other models used the same L2 parameter, learning rate, and annealing rate as the baseline model. It could be possible to further optimize each proposed model further.

### 3.4. Metrics

We report three main metrics for our networks' performance: (1) mean average precision (mAP) with respect to detection of brain lesions, (2) maximum sensitivity, and (3) segmentation DICE score of the true positives at maximum sensitivity. To derive these metrics, we must define a true positive (TP). We first binarize our 3D segmentation probability maps with probability threshold 10% and calculate the centroid of each predicted 3D CC. If the centroid is within 1 mm of any ground truth annotation, that 3D CC is treated as a TP annotation. The predicted confidence of each 3D CC is calculated as the average predicted probability from the original probability map of each voxel in the 3D CC. With the list of centroids and confidences, we create a precision-recall (PR) curve, the area under which is the mAP value. 95% confidence intervals (CIs) are reported for mAP scores using the method described by Hanley and McNeil (Hanley and McNeil, 1983). Allowing all 3D CCs after binarization to be predictions, we can also calculate our maximum sensitivity. We finally report the TP DICE scores as a segmentation metric for the lesions we do predict correctly.

## 4. Results

### 4.1. Stochastic Lesion Censoring

Table 1 shows metrics for training on stochastically censored data. With simple class-based loss weighting ($\beta = 1$), the maximum sensitivity falls to 10% of that for the baseline without censoring. After incorporating the lopsided bootstrap loss function ($\beta = 0.5$ or $\beta = 0.1$), this performance is largely recovered, up to 97% of the baseline. We also tested the network with $\alpha = 30$, which resulted in predictions of over 99% probability for every voxel regardless of the corresponding $\beta$ value.

Table 1: Stochastic Lesion Censoring with FN Rate 50%

| Training Data | Loss $(\alpha, \beta)$ | mAP (95% CI) | Max Sensitivity | TP DICE |
|---|---|---|---|---|
| Full | 3, 1 | 46 (44,47) | 80 | 72 |
| 1/2 Censored Data | 3, 1 | 20 (15,22) | 8 | 54 |
| 1/2 Censored Data | 10, 1 | 6 (2,9) | 15 | 48 |
| 1/2 Censored Data | 3, 0.5 | 39 (36,41) | 76 | 75 |
| 1/2 Censored Data | 10, 0.5 | 29 (25,32) | 53 | 69 |
| 1/2 Censored Data | 3, 0.1 | 42 (40,44) | 78 | 73 |
| 1/2 Censored Data | 10, 0.1 | 35 (31,37) | 63 | 71 |

Figure 3 shows a histogram of predicted voxel probabilities from our segmentation network on all of the test patients among our baseline model (Table 1, row 1), direct training on censored data (Table 1, row 2), and our best performing bootstrap model (Table 1, row 6). The predictions have been further colored to represent the predicted probabilities of the voxels corresponding to the ground truth (GT) lesion and GT normal classes. Thus, an ideal network would predict 0 (far left) for GT normal and 1 (far right) for GT lesion

voxels. If we renormalize the GT lesion probability histogram and calculate the Shannon entropy, we get a value of 1.74. In stark contrast, the model trained directly on the censored data without the bootstrap loss (Figure 3(b)) rarely predicts a high probability value. The entropy of our lesion probabilities is 4.38. Our bootstrap loss (Figure 3(c)) recovers some of the characteristics of our baseline model despite training on the highly censored data. The entropy of the lesion probabilities is 3.14.

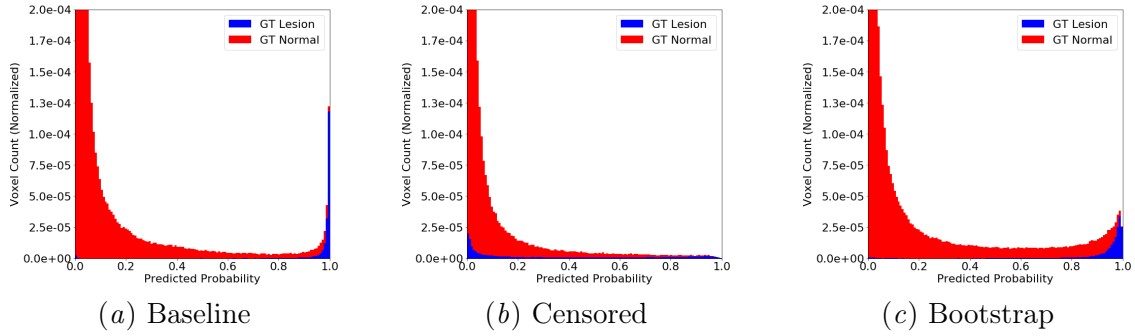

Figure 3: **Entropy of predicted probabilities.** We separate the predicted probabilities of voxels corresponding to the positive (in blue) and negative (in red) class.

Figure 4 shows representative images of the segmentations that result from training on the censored data. We can see the effects of different $\alpha$ and $\beta$ hyperparameters on our segmentations in Figure 4(a). Increasing $\alpha$ and decreasing $\beta$ both make the network more sensitive. However, decreasing $\beta$ better matches image-level gradients, which is advantageous for the task of segmentation. Figure 4(b) shows us the segmentation performance on different tiers of lesion sizes of the $(\alpha, \beta) = (3, 0.1)$ network. We can see that it performs well across a range of lesion sizes.

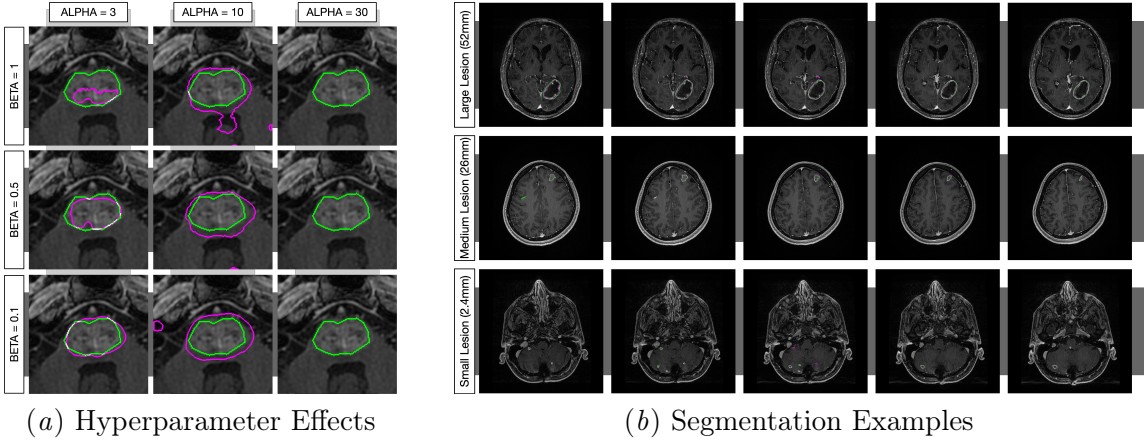

Figure 4: **Qualitative examples of segmentation results.**

### 4.2. Size-based Lesion Censoring

Table 2 shows metrics regarding our size-based lesion censoring experiments. As with stochastic censoring, training on the censored data with naive class-based loss weighting reduces performance drastically, with maximum sensitivity falling to 17% of that from the baseline with no censoring. We again demonstrate that the lopsided bootstrap loss restores performance to 88% of baseline. Similar to the experiments with random censorship, choosing $\alpha = 30$ abolished network performance.

Table 2: Size-Based Lesion Censoring with FN Rate 50%

| Training Data | Loss $(\alpha, \beta)$ | mAP (95% CI) | Max Sensitivity | TP DICE |
|---|---|---|---|---|
| Full | 3,1 | 46 (44,47) | 80 | 72 |
| 1/2 Censored Data | 3, 1 | 22 (19,24) | 14 | 61 |
| 1/2 Censored Data | 10, 1 | 18 (14,20) | 9 | 51 |
| 1/2 Censored Data | 3, 0.5 | 32 (19,34) | 68 | 71 |
| 1/2 Censored Data | 10, 0.5 | 18 (15,20) | 50 | 68 |
| 1/2 Censored Data | 3, 0.1 | 39 (37,41) | 71 | 71 |
| 1/2 Censored Data | 10, 0.1 | 19 (17,21) | 51 | 69 |

Figure 5 shows the performance with respect to lesion size of our bootstrap model trained on data with no, random, or sized-based censorship. The network trained on randomly censored data achieves better performance with smaller lesions as compared to the network trained on size-censored lesions. Indeed, the latter misses all lesions with diameter less than 4.8 mm.

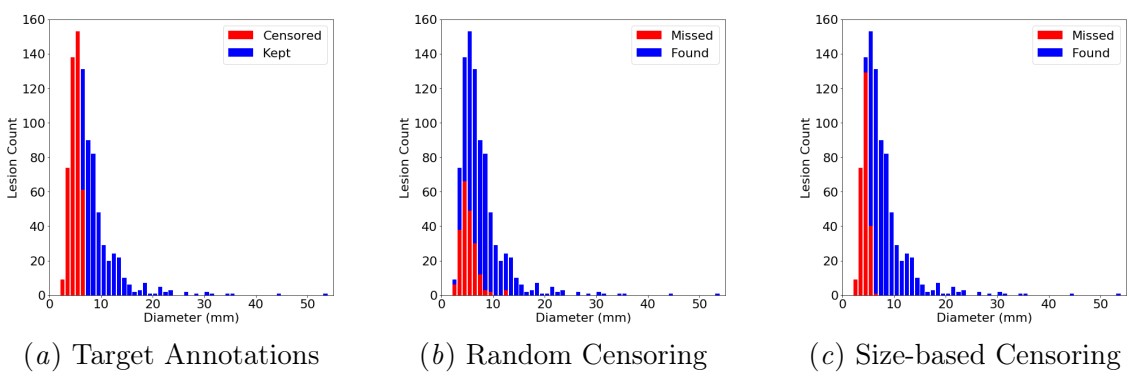

$(a)$ Target Annotations    $(b)$ Random Censoring    $(c)$ Size-based Censoring

Figure 5: **Prediction accuracy sorted by lesion size.**

### 4.3. Relationship to Patient Count

We also investigated how performance varies with dataset size by training the network on randomly subsampled patient cohorts. Training on 100 patients with a FN rate as high as

50% achieves comparable performance to training on 30 patients with fine labels. Similarly, we find similar performance training on 30 patients with noisy labels as on 10 patients with fine labels.

Table 3: Comparison of Performance with Respect to Patient Count

| Tr. Data | Pt Count | $(\alpha, \beta)$ | mAP (95% CI) | Max Sens. | TP DICE |
|----------|----------|-------------------|--------------|-----------|---------|
| Full | 10 | 3, 1 | 25 (21,27) | 44 | 63 |
| Full | 30 | 3, 1 | 39 (37,41) | 70 | 72 |
| Full | 100 | 3, 1 | 46 (44,47) | 80 | 72 |
| Stochastic | 30 | 3, 0.1 | 30 (27,32) | 54 | 69 |
| Stochastic | 100 | 3, 0.1 | 42 (40,44) | 78 | 73 |
| Size-Based | 30 | 3, 0.1 | 25 (21,27) | 48 | 69 |
| Size-Based | 100 | 3, 0.1 | 39 (37,41) | 71 | 71 |

## 5. Discussion

**Using the lopsided bootstrap loss preserves performance when training on censored data.** From Tables 1 and 2, we see that inducing a high FN rate, such as 50%, drastically reduces performance regardless of censorship model. Because of the massive voxel imbalance between the positive and negative cases, we already upweight positive pixels by a factor of 3. However, as we increase the $\alpha$ value further, we start to see a decline in overall performance, as seen in the mAP values.The lopsided bootstrap error is a better way to improve performance, as seen quantitatively in the aforementioned tables as well as qualitatively in the examples of Figure 4(a).

**The lopsided bootstrap loss can be seen as a form of entropy regularization on the positive lesion probabilities.** This can be seen most clearly by looking at the predicted probability values on our test set, as shown in Figure 3. Our baseline model shows the prediction behavior typical of most networks: predicting primarily high or low (but not intermediate) probabilities. When we censor our data, voxels of similar lesions may be annotated as either negative (normal) or positive (lesion) classes, the GT lesion voxels (Fig. 3(b)) show a more even distribution of probabilities from 0 to 1, resulting in a much higher entropy. Our bootstrap losscreates a lopsided entropy regularization by creating a positive feedback loop. This property can be detrimental, so networks should always be tested on datasets with minimum noise where possible.

**Using the bootstrap loss cannot fully abolish size-based biases.** Though performance is largely recovered in our size-based censorship experiments (Table 2, our error profiles (Figure 5(c)) show that most small lesions are missed, as the training data contains only annotated lesions larger than a certain diameter. Interestingly, our network performs better when trained with default $(\alpha, \beta)$ values given size-based censorship compared to the default network trained with stochastic censorship. One possible reason is that size-based censorship still allows the network to learn features of larger lesions without conflicting signals, since all large lesions will be labeled correctly. However, this also means that the

network actively learns that features typical of smaller lesions should be labeled as normal. As in any application of machine learning, even an optimized loss function cannot recover signals not present in the input data.

**When developing deep learning applications, consideration should be given not only to the number of samples required for the desired network performance but also the other costs of acquiring such data, such as annotator time.** Table 3 shows that our network achieves comparable performance with a larger but exceptionally noisy (FN rate of 50%) dataset as with a smaller but finely labeled dataset. Therefore, more rapidly collecting noisy data could be a beneficial tradeoff. Another application would be in mixing datasets that provide labels for different diseases that are not mutually exclusive. Our lopsided bootstrap loss function would enable training on the combined data by addressing the FNs due to each dataset potentially missing annotations for the other disease. Our work on a novel method for addressing a high prevalence of FNs in training data enables improved utilization of noisy data and complements ongoing efforts to generate more data.

**We note that this study comes with limitations.** Our simulated FN annotation does not fully simulate true clinicians' error. Additionally, 50% FN rate is too high for a true clinical simulation. However, by erring on the side of more false negatives, we hope to show the strength of our methods at the cost of accurate clinical simulation. Though small data is a limitation of this study, we believe that since learning with FN annotations presents problems independent of patient count, our main contribution (lopsided bootstrap loss) will still be useful with larger data. Finally, our original annotations not having been cross-validated among multiple readers for measurement of inter-reader variability limits our understanding of the accuracy of our non-lossy target annotations. We hope to continue to scale up our data collection process, including measuring inter-reader variability, as well as validating our method on other lesion-based datasets, such as lung nodules or liver lesions.

## 6. Conclusion

In our work, we have shown how using the lopsided bootstrap loss can help improve performance when training on a dataset whose annotations have multiple false negatives. Though the improvement is stronger when the underlying cause of the false negatives is random, it still works if the label noise happens with some bias. We hope that by creating algorithms that are more robust to noisy data and weaker labels, we can expand the domain of what annotations are usable to train deep learning networks.

## Acknowledgments

We thank Stanford Hospital for providing the data needed to complete this study. We acknowledge the T15 LM 007033 NLM Training grant in funding this project. This work was also supported in part by grants from the National Cancer Institute, National Institutes of Health, U01CA142555, 1U01CA190214, 1U01CA187947, and U01CA242879.

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
