# OpenReview forum: "Brain Metastasis Segmentation Network Trained with Robustness to Annotations with Multiple False Negatives"
_MIDL.io/2020/Conference — MIDL 2020_

### Official Review · AnonReviewer4 · 2020-03-02
**interesting work with extensive validation**

**Rating:** 4
**Confidence:** 3
**Recommendation:** Poster

**Summary:**

This paper aims to develop a method for training using noisy labels. In order to achieve this, a new loss function is developed based on entropy regularization. A simulated dataset is generated by randomly censoring lesions to create false negatives. The novel bootstrap loss function improves segmentation performance when training on data with false negatives.

**Strengths:**

-	The paper is well-structured and the case for the novel loss function is clearly outlined.
-	There is extensive validation, with examples, on the simulation and also how the results are affected by the size of the training data.


**Weaknesses:**

- the images in figure 4b are too small to evaluate - maybe these can be enlarged for the next submission.

- it is not clear from the title/abstract that the aim is to improve detection rather than segmentation.

**Detailed Comments:**

it is a bit confusing to say the max sensitivity is improved up to 97% of the baseline, it might be easier to understand to just give the sensitivity values.

**Justification Of Rating:**

This paper is interesting, novel, well-structured and the validation, while primarily based on simulation is thorough and well thought through. In addition, the limitations are clearly outlined in the discussion.

**Paper Type:**

methodological development

**Special Issue:**

no

---

> ### Author Response · Authors · 2020-03-26
> **Reply to Conference Paper48 AnonReviewer4 Comments**
>
> We thank the reviewer for the helpful comments and kind remarks.
>
> Addressing Reviewer Comments:
> - We will be moving figure 4 to its own page in the appendix for future iterations of this paper.
> - The hairy line between detection and segmentation for this paper is one that the authors wholeheartedly admit.  Though the methodology is a segmentation network, because the lesions are often quite small, the metrics for detection applied more easily.  We tried to provide both metrics to give readers the full understanding of performance of our networks.

---

### Official Review · AnonReviewer3 · 2020-03-07
**Proposes to use Bootstrap loss for for robustness against FNs**

**Rating:** 2
**Confidence:** 3
**Recommendation:** Poster

**Summary:**

 The main idea behind the paper is to make network training more robust to noisy annotations, specifically False Negatives (FNs). The paper proposes to use the bootstrap loss function to handle FN annotations. The paper is easy to follow, proposes a good method and validate it against a private decent size dataset.

**Strengths:**

+ Paper is easy to follow with clear motivation about the proposed method.
+ Method is well validation with two different types of artificial data censoring.
+ Results on the evaluated metric shows the usefulness of the proposed method.
+ Implementation details are clear and should make the paper easily reproducible if the dataset is made publicly available.
+ Limitations of the work are clearly noted and don't claim to solve all the issues with noisy annotations.

**Weaknesses:**

- There are issues regarding Equations:2 and 3.

- In the paragraph following Eq:2, it is mentioned that "$\beta=0.1$ means 90% of our loss comes from the CE between our predictions and our (potentially noisy) target annotations while 10% of our loss comes from the feedback loop of the bootstrap component." This is False as Eq:2 is $((1-\beta)*CE(y, \hat{y}) + CE(\hat{y}, argmax(\hat{y})))$ clearly $\beta=0.1$ will reduce the effect of classical cross-entropy loss (first term) to 0.9 but second term is still weighted 1.

- In the last sentence above Eq:3 it is mentioned that "With $\beta=1$, this loss simple reduces to class-based loss weighting where positive cases are unweighted by $\alpha$". This is False as $\beta=1$ will simplify Eq:3 to $(\alpha* 1[Y==1] * CE(Y,\hat{y}) + 1[Y==0] * CE(\hat{y}, argmax(\hat{y})))$. It is clear that Here, for negative class there is no classical cross-entropy loss term and it is only weighted by CE loss between prediction probability and predicted classification one-hot encoding.

- Justification for calculating Dice for only TP is necessary. Is the reported dice value on a lesion level bases or a volume level bases?

- In size based lesion censoring all metrics are reported for all lesion loads, it would be nice if these are reported for small lesions and not small lesions separately as in this experiment only small lesions were censored.

- Authors note that in size based lesion censoring of small lesions, their proposed method still misses a lot of small lesions. Can they please justify in this case what is the main benefit of the proposed method.

**Detailed Comments:**

- Can you please clarify that when you do stochastic censoring regardless of lesion size, you keep these censored "GT" same across training epochs and across different experiments?

- Can you please clarify, how the probability threshold of 10% was chosen to convert lesion segmentation to binary segmentation (Section:3.4) ?

**Justification Of Rating:**

The paper proposed an interesting approach to tackle noise in the label annotations. The Paper is well validated. But there are issues with the Equations of the proposed loss function. The evaluation metric also needs better justification.

**Paper Type:**

both

**Special Issue:**

no

---

> ### Author Response · Authors · 2020-03-26
> **Reply to Conference Paper48 AnonReviewer3 Comments**
>
> We would like to thank the reviewer for helpful and insightful comments.
>
> Addressing the Reviewer's Comments:
> - Many of the reviewer's comments deal with the inaccuracies and errors in the formulation and description of equations 1-3.  The authors fully agree with the comments and have rectified the errors as described in the "changes to equations" section below.
> - We report the average DICE score among all "found" lesions.  We limit the averaging to just the True Positives to (1) not double penalize our detection failures and (2) give a more useful metric on segmentation quality.  Because we are reporting average DICE score per lesion, having false negatives and false positives will make us take our same array of DICE scores and average them together with a 0 for each FN and FP.  However, we believed that readers would already have an accurate view of how many FN's and FP's our network outputs from the mAP and maximum sensitivity metric, and thus we only report the average DICE score for the true positive lesions.
> - Though we don't report any size-based metrics, Figure 5 does show the sensitivity of our method on different size lesions.
> - "Authors note that in size based lesion censoring of small lesions, their proposed method still misses a lot of small lesions. Can they please justify in this case what is the main benefit of the proposed method.": Though the authors acknowledge that we could not fully salvage all performance on small lesions, we do see an improvement among all metrics after training with the lopsided bootstrap loss compared to training with just weighted cross entropy, as seen in table 2.
> - "Can you please clarify that when you do stochastic censoring regardless of lesion size, you keep these censored "GT" same across training epochs and across different experiments?": We absolutely keep the censored "GT" same across training epochs and across different experiments.  The selection of the censored lesions was done once before training began, and never repeated.
> - "Can you please clarify, how the probability threshold of 10% was chosen to convert lesion segmentation to binary segmentation (Section:3.4)?": Threshold values of {3%, 10%, 30%, 50%, and 90%} were tested on the validation set.  Though this made no difference in the detection metrics (i.e. mAP and maximum sensitivity), we saw a slight increase in TP DICE score with the 10% threshold.
>
> Changes to Equations:
> Equations 2 and 3 had errors in them, and though technically faultless, equation 1 was also not fully clear.  We take a brief moment to apologize to the reviewer, as we acknowledge how frustrating it is to read bad math.  It creates unnecessary confusion and protracts the reviewing process.  This will not happen again.
> The arguments of the cross-entropy function will be flipped to match convention.  Also, I will note how each equation will be fixed below.
> Equation 1 now reads:
> $$\mathcal{L} (Y,\hat{Y}) = \left\{\begin{array}{lc}
> CE(Y,\hat{Y})&\text{if } Y == 0\\
> \alpha * CE(Y,\hat{Y})&\text{if } Y == 1
> \end{array}
> \right.$$
> We hope that splitting the equation to a conditional provides further clarity on the weighting of the positive case.
> Equation 2 now reads:
> $$\mathcal{L}(Y,\hat{Y})= \beta * CE(Y,\hat{Y})+ (1-\beta)*CE(\text{argmax}(\hat{Y}),\hat{Y}))$$
> In addition, the paragraph explaining it has changed to describe the correct nature of $\beta$.  This makes equation 2 fit in with equation 3 as well as all the tables of experiments.
> Equation 3 now reads:
> $$\mathcal{L} (Y,\hat{Y}) = \left\{\begin{array}{lc}
> \beta * CE(Y,\hat{Y})+ (1-\beta)*CE(\text{argmax}(\hat{Y}),\hat{Y}))&\text{if } Y == 0\\
> \alpha * CE(Y,\hat{Y})&\text{if } Y == 1
> \end{array}
> \right.$$
> We hope that this shows a clearer combination of ideas between equation 1 and equation 2.

---

### Official Review · AnonReviewer1 · 2020-03-10
**A new network for Brain Metastasis Segmentation**

**Rating:** 4
**Confidence:** 3
**Recommendation:** Oral

**Summary:**

This paper tries to describe a new method for brain metastasis segmentation using Network Trained with Robustness to Annotations with Multiple False Negatives. The novel contribution of authors in this paper is:
1. segmentation network with whole -lesion false negative labels.
2. this method preserves performance for high induced FN rates (as much as 50%)

As the authors specified in their paper their network can overcome to the error segmentation that introduce to segmentation in two ways:
1. Stochastic Censoring: that some error segmentation is induced to true label segmentation
2. Size-Based Censoring: by censoring the smallest lesions (by volume) across all patients

Their model was able to overcome these errors with high performance (98%)

**Strengths:**

This is very well written paper with a new method and the results are eye-catching.
The material and methods have been described well.
The result section is clear and figures describe their work properly.
The discussion section is complete and describes the methods that was used in this paper.

**Weaknesses:**

The number of cases is not too much and are not from multiple sites.
Equation 1, 2 and 3 are somewhat dumb and need more definitions.
The  original annotations not having been validated among multiple readers for measurement of inter-reader variability.
The affect of different scanners on the performance of their model was not assessed.

**Justification Of Rating:**

For Goals, fully achieved; i.e. the authors accomplished their results and target associated with the particular goal being rated.
For Performance Factors, results met all standards, expectations, and objectives.
For Overall Performance, expectations were consistently met and the quality of work overall was very good.

**Paper Type:**

both

**Questions To Address In The Rebuttal:**

Equation 1 and 2 are somewhat dumb and need more definitions.

**Special Issue:**

yes

---

> ### Author Response · Authors · 2020-03-26
> **Reply to Conference Paper48 AnonReviewer1 Comments**
>
> We thank the reviewer for the detailed comments and useful suggestions for improvement.
>
> Questions to Address in the Rebuttal:
> All three equations have created confusion for all reviewers and will be heavily addressed.  We have written a section going into detail of how the equations will be changed below.
>
> Changes to Equations:
> Equations 2 and 3 had errors in them, and though technically faultless, equation 1 was also not fully clear.  We take a brief moment to apologize to the reviewer, as we acknowledge how frustrating it is to read bad math.  It creates unnecessary confusion and protracts the reviewing process.  This will not happen again.
> The arguments of the cross-entropy function will be flipped to match convention.  Also, I will note how each equation will be fixed below.
> Equation 1 now reads:
> $$\mathcal{L} (Y,\hat{Y}) = \left\{\begin{array}{lc}
> CE(Y,\hat{Y})&\text{if } Y == 0\\
> \alpha * CE(Y,\hat{Y})&\text{if } Y == 1
> \end{array}
> \right.$$
> We hope that splitting the equation to a conditional provides further clarity on the weighting of the positive case.
> Equation 2 now reads:
> $$\mathcal{L}(Y,\hat{Y})= \beta * CE(Y,\hat{Y})+ (1-\beta)*CE(\text{argmax}(\hat{Y}),\hat{Y}))$$
> In addition, the paragraph explaining it has changed to describe the correct nature of $\beta$.  This makes equation 2 fit in with equation 3 as well as all the tables of experiments.
> Equation 3 now reads:
> $$\mathcal{L} (Y,\hat{Y}) = \left\{\begin{array}{lc}
> \beta * CE(Y,\hat{Y})+ (1-\beta)*CE(\text{argmax}(\hat{Y}),\hat{Y}))&\text{if } Y == 0\\
> \alpha * CE(Y,\hat{Y})&\text{if } Y == 1
> \end{array}
> \right.$$
> We hope that this shows a clearer combination of ideas between equation 1 and equation 2.

---

### Official Review · AnonReviewer2 · 2020-03-12
**A loss to increase robustness against FN, but lacks proper experimental validation**

**Rating:** 2
**Confidence:** 4

**Summary:**

This paper presents an approach to regularize the training of a segmentation model to reduce the impact of false negatives in the training set. The proposed loss gives more weight to the positive class while penalizing the entropy of the predictions on the negative class. The evaluation is performed on a recently published dataset for brain metastasis segmentation.


**Strengths:**

- The paper tackles an interesting problem related to practical issues of human errors and labeling times for medical image segmentation.
- The problem is well defined, justified and presented, and it feels like the authors know well the medical aspects.

**Weaknesses:**

- This paper misses on a large part of the literature on noisy label learning. This problem has a significant number of publications every year that are not discussed at all here.
- The formulation of the loss is not intuitive nor explained clearly enough, with imprecise notation (see detailed comments section).
- The experimental protocol is not thorough: the hyper-parameters of the loss are tested within very limited ranges and miss key values (1 for $\alpha$ and 0 for $\beta$); the censoring is not explained properly as it is not clear if the data is censored randomly at the start of the training or censored differently for every iteration (which would make a significant difference in the stochastic case).
- This work does not compare with any baseline: a (well tuned) cross-entropy baseline would have been expected in the tables. I feel this is an important missing point of this paper. What is called baseline in the abstract is not a baseline.
- The conclusions are either very general (e.g. "When developing deep learning applications, consideration should be given not only to the number of samples required for the desired network performance but also the other costs of acquiring such data, such as annotator time.") or not well supported claims (e.g. "Using the bootstrap loss cannot fully abolish size-based biases."). This makes the conclusions sound more like speculation rather than demonstration.

**Detailed Comments:**

- Section 2 on how the data was acquired is not necessary and breaks anonymity (such details can only come from the authors of the article for that dataset).
- Number of censoring $2^n$ seems incorrect. If you censor $k$ (with $k = p * n$) lesions out of $n$, you have ${n}\choose{k}$ combinations.
- There seems to be an error in equation 2: the $\beta$ weight seems to be missing from the second part of the equation: $\mathcal{L}(\hat{Y},Y)=(1-\beta)CE(\hat{Y},Y)+\beta (\hat{Y},\mathrm{argmax}\hat{Y})$
- Notations:
   + Notations between equation 1, 2 and 3 needs to be unified. It is not clear if we are talking about the same loss but combined or if it's different formulations since there is a missing $\beta$ in equation 2 and the $1-\alpha$ factor is changed to $\alpha$.
   + In equations 1 and 3, the class testing term $\mathbf{1}[Y==1]$ is an unconventional notation. Iverson brackets should be used here with only one operator as such: $[Y=1]$. Writing with two operators is a (language specific) programming convention. Also, there is no need for $\mathbf{1}$ in front as this is the indicator function which indicates the membership of an element to a set.
   + CE refers to "the common cross-entropy" as stated in the paper, but usually the first argument of CE is $Y$ and the second is $\hat{Y}$. With common convention on the notation of CE, we would have $CE(\hat{Y},Y) = -\sum_k \hat{y}_k \log(y_k)$ which is undefined.
- "With β = 1, this loss simply reduces to class-based loss weighting where positive cases are upweighted by α." This is not true: with $\beta = 1$, the loss reduces to $\mathcal{L}(\hat{Y}, Y) = \alpha[Y=1]CE(\hat{Y},Y)+[Y=0]CE(\hat{Y},\mathrm{argmax}\hat{Y})$. This is true when $\beta=0$.
- The range of tested values for $\alpha$ is quite small. I would at least expect to see 1 in it. Same for $beta$, testing 0 seems important here (if there is no mistake in the equations).
- "The validation set had the same lesion censorship as the training set". Does it mean that evaluating on the validation set when training with stochastic censorship provides different results if run several times? This is a very unusual practice.
- "We also tested the network with $\alpha = 30$, which resulted in predictions of over 99% probability for every voxel regardless of the corresponding $\beta$ value." This is not surpising at all: with $\alpha=30$, the CE on the positive class completely dominates in the optimization, making the network predict everything positive.
- Dice is reported only on correctly predicted lesions. Does this mean that if the network misses a lesion, the Dice will not take it into account? This is a very unusual practice.
- The entropy of the distribution of the probabilities is calculated but misses an interpretation.
- Figure 3. is wrongly dubbed "Entropy of predicted probabilities." but this figures represents histograms.
- "The lopsided bootstrap loss can be seen as a form of entropy regularization on the positive lesion probabilities." This is obvious from the term $CE(\hat{Y}, \mathrm{argmax}\hat{Y})$ which is exactly entropy regularization on the predicted probabilities.
- Figure 4. could be put in appendix on a full page as it is too small right now to be seen when printed.

**Justification Of Rating:**

Overall, this paper presents an interesting problem but lacks the rigor in the presentation of the proposed solution (notations, explanations) and the experiments are not thorough enough. The dataset used to evaluate seems to not be publicly available, but the authors did not put effort into evaluating against a decent baseline.

**Paper Type:**

methodological development

**Questions To Address In The Rebuttal:**

1. What are the performances of a baseline model trained with cross-entropy only? This question can be extended to focal loss, generalized dice loss and boundary loss.
2. Is the formulation of the loss in the paper correct? (equation 3). From the text it seems that this is not the case.
3. How was the stochastic censoring applied? If lesions are randomly removed at each training iteration, it is not surprising to see that this obtains a good performance as this would correspond to a form of data augmentation, and would not correspond to adding FNs in the dataset.

**Special Issue:**

no

---

> ### Author Response · Authors · 2020-03-26
> **Reply to Conference Paper48 AnonReviewer2 Comments**
>
> We thank the reviewer for the meticulous reading of our work and helpful suggestions for improvement.
>
> Questions to Address in the Rebuttal:
> 1. "What are the performances of a baseline model trained with cross-entropy only?": The model trained with only cross entropy performs significantly poorer than our $\alpha=3$ baseline model.  The cross-entropy-only (e.g. $\alpha=1$) model has the following metrics: mAP = 0.35, max sensitivity = 0.68, and true-positive DICE score 0.69.  When comparing the lopsided bootstrap model performance, we wanted to compare with the optimal baseline that does not use the bootstrap loss.  This is why we chose to call our $\alpha=3$ model our baseline model.  However, the reviewer has highlighted the importance of the true baseline model as well, and we will include these metrics in the revised write-up if given the chance.  However, we did not test and do not have the time to run further metrics on other losses mentioned by the reviewer, such as focal loss, generalized dice loss, and boundary loss.
> 2. "Is the formulation of the loss in the paper correct? (equation 3). From the text it seems that this is not the case.": The reviewer is correct.  Equations 2 and 3 had errors in them, which will be corrected as detailed in the "changes to equations" section below.
> 3. "How was the stochastic censoring applied? If lesions are randomly removed at each training iteration, it is not surprising to see that this obtains a good performance as this would correspond to a form of data augmentation, and would not correspond to adding FNs in the dataset.": The stochastic censoring was applied ONCE before the training was started.  Thus, we trained our network on a single pool of censored data whose censoring was determined before training ever happened.  Thus, each iteration, the network sees the same version of censoring, and not a different censoring.  If it were different, the reviewer would be 100% correct in noting that this would be a data augmentation technique.  The authors acknowledge that this must be made clearer in the revision.
>
> Changes to Equations:
> Equations 2 and 3 had errors in them, and though technically faultless, equation 1 was also not fully clear.  We take a brief moment to apologize to the reviewer, as we acknowledge how frustrating it is to read bad math.  It creates unnecessary confusion and protracts the reviewing process.  This will not happen again.
> The arguments of the cross-entropy function will be flipped to match convention.  Also, I will note how each equation will be fixed below.
> Equation 1 now reads:
> $$\mathcal{L} (Y,\hat{Y}) = \left\{\begin{array}{lc}
> CE(Y,\hat{Y})&\text{if } Y == 0\\
> \alpha * CE(Y,\hat{Y})&\text{if } Y == 1
> \end{array}
> \right.$$
> We hope that splitting the equation to a conditional provides further clarity on the weighting of the positive case.
> Equation 2 now reads:
> $$\mathcal{L}(Y,\hat{Y})= \beta * CE(Y,\hat{Y})+ (1-\beta)*CE(\text{argmax}(\hat{Y}),\hat{Y}))$$
> In addition, the paragraph explaining it has changed to describe the correct nature of $\beta$.  This makes equation 2 fit in with equation 3 as well as all the tables of experiments.
> Equation 3 now reads:
> $$\mathcal{L} (Y,\hat{Y}) = \left\{\begin{array}{lc}
> \beta * CE(Y,\hat{Y})+ (1-\beta)*CE(\text{argmax}(\hat{Y}),\hat{Y}))&\text{if } Y == 0\\
> \alpha * CE(Y,\hat{Y})&\text{if } Y == 1
> \end{array}
> \right. $$
> We hope that this shows a clearer combination of ideas between equation 1 and equation 2.
>
> Additional Comments from Authors:
> - The number of possible censored datasets is wrong in the original manuscript.  It will be updated to use combinations and not the exponent.
> - We report the DICE score on only the "found'' lesions to (1) not double penalize our detection failures and (2) give a more useful metric on segmentation quality.  Because we are reporting average DICE score per lesion, having false negatives and false positives will make us take our same array of DICE scores and average them together with a 0 for each FN and FP.  However, we believed that readers would already have an accurate view of how many FN's and FP's our network outputs from the mAP and maximum sensitivity metric, and thus we only report the average DICE score for the true positive lesions.
> - We will be moving figure 4 into the Appendix and resizing it to take up a full page, for more visual clarity.

---

### Meta-Review · Area_Chair1 · 2020-04-06
**MetaReview of Paper48 by AreaChair1**

**Rating:** 3
**Recommendation For Accepted Papers:** Poster

**Metareview:**

The reviews consistently emphasize that the paper is focused and mostly well-written (some concerns about equations clarity are addressed by the rebuttal). I agree with reviewer 1 about the usefulness of the base case alpha=1, beta=0 (for completeness). The authors provided such numbers in the rebuttal and should include these into the paper.

As noted by reviewer 4, the authors should clearly emphasize that the paper aims at improving detection, rather than segmentation. I strongly encourage the authors to revised the title.

For completeness, I would also like to encourage the authors to add technical explanation (even though it is standard) on relationship between prediction entropy and loss (2) using argmax. This would further improve readability.

The methodological novelty of the paper is relatively minor (empirical study of re-weighting of standard terms), which explains the poster rating.

**Paper Type:**

validation/application paper

**Special Issue:**

no

---

### Decision · Program_Chairs · 2020-04-11

Accept